# State-of-the-Art of Profiling Immune Contexture in the Era of Multiplexed Staining and Digital Analysis to Study Paraffin Tumor Tissues

**DOI:** 10.3390/cancers11020247

**Published:** 2019-02-20

**Authors:** Edwin Roger Parra, Alejandro Francisco-Cruz, Ignacio Ivan Wistuba

**Affiliations:** Department of Translational Molecular Pathology, The University of Texas MD Anderson Cancer Center, 2130 West Holcombe Blvd, Houston, TX 77030, USA; AFrancisco@mdanderson.org (A.F.-C.); iiwistuba@mdanderson.org (I.I.W.)

**Keywords:** immune profiling, cancer tissues, multiplexed methodologies, image analysis, spatial analysis

## Abstract

Multiplexed platforms for multiple epitope detection have emerged in the last years as very powerful tools to study tumor tissues. These revolutionary technologies provide important visual techniques for tumor examination in formalin-fixed paraffin-embedded specimens to improve the understanding of the tumor microenvironment, promote new treatment discoveries, aid in cancer prevention, as well as allowing translational studies to be carried out. The aim of this review is to highlight the more recent methodologies that use multiplexed staining to study simultaneous protein identification in formalin-fixed paraffin-embedded tumor tissues for immune profiling, clinical research, and potential translational analysis. New multiplexed methodologies, which permit the identification of several proteins at the same time in one single tissue section, have been developed in recent years with the ability to study different cell populations, cells by cells, and their spatial distribution in different tumor specimens including whole sections, core needle biopsies, and tissue microarrays. Multiplexed technologies associated with image analysis software can be performed with a high-quality throughput assay to study cancer specimens and are important tools for new discoveries. The different multiplexed technologies described in this review have shown their utility in the study of cancer tissues and their advantages for translational research studies and application in cancer prevention and treatments.

## 1. Introduction

Despite the recent advances in immunotherapy strategies in recent years in cancer treatment and clinical responses, the study of immune cell phenotypes and their spatial distribution at the tumor site has prompted the need for multiplexed analyses of tumor tissues. To address this necessity, in recent years, multiplexed imaging platforms have arisen as important tools that can provide critical information about the cancer microenvironment, prognosis, therapy, and relapse [1,2,3,4,5]. Different components of the tumor microenvironment can be examined simultaneously using multiplexed methodologies, providing an insight into the biological cross-talk present at the tumor–host interface, and providing information from the subcellular level to the cell population level. Indeed, the most important factor is the precision with these new techniques can evaluate the special localization of multiple, simultaneously-detected biomarkers and their co-expressions or interactions between cells [5]. Attempts are presently being made to develop even more comprehensive multiplexed technologies that allow simultaneous visualization of an even larger number of biomarkers from a single tissue section, as well as to streamline, automate, and reduce the time expended on tissue staining and processing. Multiplexed methods can help to achieve these technological goals to ultimately enhance disease diagnosis and better inform timely patient care [6].

Multiplexed technologies are being used to identify the presence of multiple biological markers on a single tissue sample or an ensemble of different tissue samples [7]. The multiplexed imagining techniques provide unique biological information that, in many cases, cannot be attained by other non-imaging methods or by single immunohistochemistry (IHC) techniques. As mentioned, individual cells can be accessed with extraordinary fidelity equal to that achievable in the bulk population, such than even rare cell populations can be studied, showing their important role in translational research. This knowledge can be applied in cancer prevention. In this review, we discuss the most recent multiplexed methodologies that can be used to identify simultaneous biomarkers in formalin-fixed, paraffin-embedded (FFPE) tumor tissue samples as well as imaging analysis platforms with potential application for future cancer immunotherapy biomarker discoveries.

## 2. Non-Fluorescence-Based Platforms

### 2.1. Multiplexed Immunohistochemical Consecutive Staining on Single Slide

The multiplexed immunohistochemical consecutive staining on single slide (MICSSS) [8] method is a series of sequential cycles of staining, image scanning, and destaining of chromogenic substrate than can be performed on FFPE tissue samples. This multiplex staining approach uses conventional chromogenic-immunohistochemistry staining, followed by a scanning process by the destained chromogenic substrate in organic solvent [8] that can completely remove the staining. The MICSS method can allow up to 10 different antibodies on one single tissue section using sequential cycles without any damage to the tissue antigenicity or architecture. The relatively slow process of the technique is the main limitation of the MICSSS, but as the authors mentioned, this limitation could be easily resolved with the automation of the process. However, although this methodology was tested on limited data, it showed the versatility and potential of the process to study and analyze the complexity of the tumor microenvironment.

### 2.2. Sequential Immunoperoxidase Labeling and Erasing

Sequential immunoperoxidase labeling and erasing (SIMPLE) is a multiplex immuno-histochemistry approach with a sequential labeling bleaching technique that enables simultaneous marker visualization [9]. The SIMPLE approach can combine five to twelve markers using the alcohol-soluble peroxidase substrate 3-amino-9-ethylcarbazole with a fast, non-destructive method for antibody–antigen separation. Then, in each round of labeling, a given precipitate is gave a pseudocolor, and all colors are overlapped at the end of the process to visualize all of the target antigens used. This method has shown the ability to erase the results of a single stain while preserving tissue antigenicity for repeated rounds of labeling [9]. Using the SIMPLE platform in a head and neck squamous cell carcinoma cohort, differential immune complexity of lymphoid- and myeloid-inflamed tumors has been demonstrated, correlating with clinical outcomes and tumor subclassification. In addition, geometrical mapping analysis revealed that the immune complexity status is associated with the therapeutic response to vaccination therapy in pancreatic ductal adenocarcinoma, where myeloid-inflamed and T cell exhaustion status are correlated with a shorter overall survival time [10].

## 3. Fluorescence-Based Platforms

### 3.1. Bleaching Techniques without Signal Amplification System

Multiplexed staining bleaching techniques were created with different platforms to study tumor tissue specimens. The basic concept of these techniques is to erase the staining marker when it is done to initiate the next biomarker in a consecutive cycle of desired biomarkers to identify multiple antigens in a single sample.

### 3.2. Multi-Epitope-Ligand Cartography

Multi-epitope-ligand cartography (MELC) [11,12], is a bleaching or erasure technique that is capable of co-localizing the locations of different proteins in one single tissue sample using consecutives rounds of conjugate biomarkers with fluorescent detection [13]. A couple of antibodies are added during each staining cycle, followed by image acquisition of the sample using a high-sensitivity digital camera. Then, the sample is bleached with phosphate buffer saline to eliminate the excitation wavelengths, and a new cycle of staining is started. One limitation of the MELC technique is that the photobleaching step can only be applied to the microscope’s field of view, meaning that the multiprobe image is limited to a single microscopic medium-to-high power field [9]. MELC can be applied on FFPE and frozen tissue sections, and it has been coupled to RNA extraction to combine RNA and protein expression analysis [14]. MELC has been used as a very efficient methodology to study immune cell markers and intracellular signaling pathways in an infectious context [15,16] and to perform systematic high-content proteomic analysis of colorectal cancer, and the T cell-related protein expression patterns and their modification in the tissue of Barrett’s esophagus and esophageal adenocarcinoma patients [17,18].

### 3.3. MultiOmyx^TM^ Staining or Hyperplexed Immunofluorescence Assay

General Electric Healthcare (Niskayuna, NY, USA) has developed an erase methodology platform called MultiOmyx^TM^, which is a multiplex direct immunofluorescence approach where up to 50 antibodies can be interrogated from a single FFPE section. It uses primary conjugated antibodies with fluorochromes to stain different biomarkers of interest in batches of two or four at one time. After deactivating the tissue autofluorescence and completing the first cycle of staining, the tissue is imaged and deactivation of the fluorochromes via alkaline oxidation is done to start a new cycle of staining. The MultiOmyx^TM^ platform can stain multiplex rounds of biomarkers by repeating the same procedure several times until all desired targets have been reached in a multiplexed iterative manner [19,20]. The MultiOmyx^TM^ platform has been used to evaluate the epithelial-to-mesenchymal transition in medullary colorectal cancer tissue where coexpression of CK, CDH3, VIM, and Cyt-PLAC8 provided evidence that excess PLAC8 is involved in the epithelial-to-mesenchymal transition [21]. An interesting field of application of this technology is in hematopathology where the routine diagnosis of hematological neoplasms includes several IHC markers, for example, CD30, CD15, PAX-5, CD20, CD79a, CD45, BOB.1, OCT-2, and CD3 antibodies in the diagnosis of classical Hodgkin lymphoma. It was demonstrated that the use of MultiOmyx^TM^ is equivalent to routine morphological and IHC evaluation of cases in which classical Hodgkin lymphoma was included within the differential diagnoses [12].

## 4. Tissue-Based Cyclic Immunofluorescence (t-CyCIF) Method

Recently described in the literature, the tissue-based cyclic immunofluorescence (t-CyCIF) [22] method can create highly multiplexed images using an iterative process in which conventional low-plex fluorescence images are repeatedly collected from the same sample and then assembled into a high-dimensional representation. The t-CyCIF cycles involve antibody staining against protein antigens, nuclear staining (same fluorophore per cycle), image scanning (low and high magnification) and fluorophore bleaching steps. According to the authors, the cycles can be repeated more than 15 times without any problem with cell preservation or tissue morphology to complete all the desired targets. However, each t-CyCIF cycle involves a relatively slow process (each cycle is 6–8 h); a single operator can process 30 slides in parallel with relative flexibility. Recently, Bolognesi and colleagues [23] described multiplex staining by sequential immunostaining involving 30 markers using beta-mercaptoethanol/sodium dodecyl sulfate for the stripping procedure during each cycle of staining and scanning with very good results.

### Co-Detection by Indexing or Fluorescent Immunohisto-PCR

CO-Detection by indEXing (CODEX) is a fluorescent-based imaging approach that uses oligo-DNA conjugated antibodies. The oligonucleotide duplexes encodes uniquely designed sequences with 5’ overhangs [24]. Fresh frozen tissue and isolated cells were used to validate this methodology, but its application on FFPE tissue is under development. Cells or fresh frozen tissue are stained with a cocktail containing all conjugated antibodies (up to 50 antibodies) at the same time. This methodology is based on secondary detection index cycles where tags are iteratively revealed in situ by using indexing nucleotides (adenine and guanine) and rendering fluorophores-conjugated nucleotides (uracil and cytidine) with the combination of a polymerization cycle and a fluorescent channel, at which a given DNA tag incorporates one of two fluorescently labeled dNTP species. Specifically, the antibody-matched overhangs (indexes) include a region to be filled by blank letters and a dedicated position for a dye labeled nucleotide at the end. The antibodies to be revealed first generally have shorter overhangs than the antibodies to be visualized later (Figure 1). Each extension and bleaching (with TCEP) cycle takes 10 min. Imaging in each cycle takes min to hours depending on sample dimensions, resolution, and the microscope used (a standard fluorescence microscope). The platform can be performed on any three-color fluorescence microscope enabling the conversion of a regular fluorescence microscope into a tool for multidimensional tissue rendering and cell cytometry [24], giving a good advantage to users of this platform. CODEX is an innovate platform that has achieved and reported deep immune profiling of the mouse splenic architecture by comparing normal murine spleens to spleens from animals with systemic autoimmune disease [14]. Another barcoding platform is the DNA exchange imaging (DEI) technique [25] that overcomes speed restrictions by allowing for single-round immunostaining with DNA-barcoded antibodies. The DEI is the new generation of exchange-PAINT described by the same group [26]. According to the authors, it is an easy multiplexed technique that can be adapted to diverse imaging platforms, including standard resolution Exchange-Confocal and various super-resolution methods. There are no cancer-related study publications using these methods, but they are promising techniques and highly efficient methodology to study the tumor-associated immune contexture.

## 5. Amplification of the Epitope Detection

### 5.1. Multiplex Modified Hapten-Based Technology

Modified-hapten based technology is a recent technique that allows simultaneous detection of multiplex biomarkers using a standard two-step procedure. The technique is antibody species independent and the signals of the markers can be stronger than those usually observed with direct flour-labeled secondary antibody detection of multiplex. Created by the company Cell IDx (San Diego, CA, USA), primary antibodies are combined in cocktails and then detected with a panel of anti-hapten secondary antibodies, each labeled with a different fluorochrome. The procedure takes two hours [27], which is a principal advantage of this multiplexed method.

### 5.2. Tyramide Signal Amplification and Fluorescent Multiplex Immunohistochemistry

Tyramide signal amplification (TSA) was described in the 1990s by Bobrow and colleagues [28,29]. It is an enzyme-linked signal amplification method that is using to detect and localize the low copy number of proteins present in tissue by the conventional IHC protocol, using, most commonly, the alkaline phosphatase or horseradish peroxidase (HRP) enzymatic reaction to catalyse the deposition of tyramide labelled molecules at the site of the probe or epitope detection. Tyramides are conjugated to biotin or fluorescent labels and revealed by the streptavidin–HRP system [6,30]. The HRP catalyzes the formation of tyramide into highly reactive tyramide radicals that covalently bind to electron-rich tyrosine moieties close to the epitope of interest on FFPE tissue. Tissue surfaces with anchored biotinylated tyramide must be further treated with fluorescent or enzyme tagged proteins that have a high affinity for biotin, such as streptavidin, before microscopic visualization [6,30]. The detection of the proteins is more than 10-times greater than standard biotin-based staining methods [31].

Akoya/PerkinElmer (Waltham, MA, USA) developed the Opal^™^ workflow (Figure 1), which allows simultaneous staining of multiple biomarkers within a single paraffin tissue section. Fluorescent Multiplex Immunohistochemistry (fmIHC) allows researchers to use antibodies raised in the same species, and different panels combining different targets can be created using this technology [4,30]. The manual protocol approach involves detection with fluorescent TSA reagents, followed by microwave treatment that removes the primary and secondary antibodies between cycles and any nonspecific staining that reduces tissue autofluorescence for each antibody cycle. In the automated protocol using Leica Bond RX or another autostainer, the time is reduced drastically as compared with manual staining. The possibilities for fmIHC are expanding our knowledge of tumor immune contexture. Mapping the tumor microenvironment and the predictive and/or prognostic value of immune checkpoint expression on malignant cells and tumor infiltrating immune cells has been characterized in patients with melanoma, lung cancer, breast cancer, gastric cancer, Hodgkin lymphoma, and others by fmIHC [32,33,34,35,36].

### 5.3. Nanocrystal Quantum Dots

The method uses specially coated nanocrystals (around 1–10 nm in diameter) called quantum dots instead of the chromogen [37,38]. Nanocrystal quantum dots have the property of being excited by any type or wavelength of light to emit light in a very thin fluorescence spectrum. The use of these fluorescent markers in combination with multispectral imaging technology has been a particular utility for multiplexed detection when used as a fluorescent probe bound to different antibody markers [39,40]. Despite the favorable optical properties of nanocrystal quantum dots, as a fluorescence-based method, they can avoid the endogenous autofluorescence associated with tissue sections [41], have high photostability [42], and have a symmetric emission spectrum [43]. An important reported limitation of using nanocrystal quantum dots is the limited number of nanocrystals that possess the proper chemistry to attach themselves to their targeted molecules. Nanotechnology is a promising platform in cancer nanodiagnostics and nanotherapy because of the unique optical and electronic features. When conjugated with antibodies, QD-based probes can be used to target cancer molecules with high specificity and sensitivity [36]. In addition, the use of QD-based multifunctional probes has been proposed for multiplexed molecular cancer diagnosis, and in vivo imaging [36,44].

## 6. Fundamentals of Multiplexed Techniques Based on Mass Spectrometry

### 6.1. Imaging Mass Spectrometry

Imaging mass spectrometry (IMS) is defined as the visual representation of the elemental or molecular component of fixed cells or tissues by mass spectrometry (Figure 2) [27]. IMS is a technique that uses a mass spectrometer (MS) to visualize the spatial distribution of compounds, biomarkers, metabolites, proteins, peptides, or small molecules by their molecular masses [45]. The incorporation of a computer data system to mass spectrometry started the path of IMS. In 1967, two computational systems were applied to MS [28]. The Massachusetts Institute of Technology system was the first computer-assisted digital data acquisition system for this purpose. The software identifies the mass spectral peaks and assigns them mass values and intensities to transform the results in numerical and graphical form [46]. The Stanford system was the second system created. It uses computer software that controls the data acquisition from a quadrupole MS interfaced to a gas chromatograph that scans each spectrum from peak to peak each spectrum to predetermine the total ion current [46].

IMS is applied to biological and non-biological samples such as cells, tissues, polymers, and minerals [47,48,49]. In general, the methodology can be applied to all systems mentioned; it analyzes a thin section of the sample placed on a target-plate. The sample is introduced into the source region of the MS where the surface is subjected to bombarding ions, photons, and/or atomic or molecular beams. Then, compounds present in the sample are ionized and mass analyzed. This process is then repeated as necessary in a raster over a selected region of interest until the complete desired area has been sampled. The intensity of any given ion may be plotted as a function of its *x* and *y* positions, thus generating specific two-dimensional molecular/ion images of the sample [46].

### 6.2. Secondary Ion Mass Spectrometry

Secondary Ion Mass Spectrometry (SIMS) achieves chemical or elemental analysis of surface constituents, rather than being excited to emit some characteristic secondary signals, such as fluorescence-based techniques. A light (static SIMS) or heavy (dynamic SIMS) energetic primary ion causes a collision cascade. Those ionized particles from the surface are subsequently identified with MS [46,50]. This is one of the most destructive methods of surface analysis, but the it is the most sensitive as all elements are detectable, including hydrogen [50]. During the analysis, the sample surface is gradually eroded away [50].

SIMS was developed for the elemental analysis of non-biological and biological surfaces, such as the study of a lunar basalt from the Apolo 11 in the 1970s and the study of the insect abdomen tissue morphology in 1975 [49,51]. The sources of the primary ion beam that have been used more frequently are Ar^+^, O_2_^+^, N^+^, O^−^, and Cs^+^; however, in principle, the primary ion may be a positive or negative ion. Noble or reactive gas ions are usually extracted from discharged plasma. The secondary ion beam detection includes stable and radio isotopes like ^2^D, ^15^N, ^13^C, ^18^O, ^33^S, ^74^Se ^90^Zr, ^56^Fe, ^40^Ca, and ^14^C. Since 1960, there have been two ways to acquire the secondary ion content: an ion microscope instrument mode (Cameca, Gennevilliers, Paris, France) and a scanner ion microprobe mass analyzer (IMMA) mode (Applied Research Laboratories, Austin, TX, USA) [52]. In microscope mode, the MS analyzes one ion per time, and its position is mirrored on the detector, and each ion image is generated independently. In the microprobe mode, the primary ion beam rasters the sample to produce a mass spectrum at small locations on the sample surface. An entire mass spectrum is obtained at each position or pixel, and the images may be constructed by plotting the ion intensities across the sample in a two-dimensional fashion [46].

SIMS has been compared with electron microscopy because of the resolution of its images; however, an ideal instrument should have a lateral resolution of 100 A, a mass resolution for secondary ions better than 10,000 A, a secondary ion transmission of close to 100%, and simultaneous detection of all secondary ions [50].

Time-of-Flight (TOF) Secondary Ion Mass Spectrometry started to be applied to biological cells as a chemometric methodology to study the cellular surface composition and the discriminations between normal and neoplastic cells, an issue that can be challenging in cases where neoplastic morphological features may not be evident, such as low grade prostate cancer and bladder cancer [53,54], or to study the chemical composition that can differentiate subtypes of well-defined neoplasia, such as estrogen-receptor-positive (ER+) and estrogen-receptor-negative (ER−) breast cancers [55].

### 6.3. Laser Desorption/Ionization

Laser desorption/ionization (LDI) is another IMS platform created in the 1960s that nebulizes a solid surface in order to obtain free ions or ion clusters for imaging. It involves the use of lasers, UV or IR, instead of an ion beam. The coupling of LDI to Time-Of-Flight (TOF) mass analyzer was possible in the 1970s, and the first report of metal bioimaging by laser ablation inductively coupled plasma mass spectrometry (LA-ICP) was reported in 2010 to have high sensitivity and quantitative abilities. One of the first biological reports of these methodologies involved the quantitative imaging of copper in the human hippocampus and substantia nigra [46]. One of the most interesting applications of this platform is to test the efficacy of metal-based anticancer agents into tumor models, such as the distribution of platinum-based anticancer compounds in a human colorectal cancer spheroid model [56].

### 6.4. Matrix-Assisted Laser Desorption/Ionization

Matrix-Assisted Laser Desorption/Ionization (MALDI), a type of molecular imaging technology, was evolving in the same way as SIMS and LDI, with improved resolution and sensitivity. MALDI imaging initially applied TOF-MS, but other platforms were coupled to MALDI later on with the objective of localizing small pharmaceutical molecules directly on tissue sections [46,57]. MALDI is a soft ionization technique that uses an organic compound matrix such as 2.5-dihydroxy benzoic acid (DHB) that, when combined with pulsed UV or N_2_ laser irradiation, promotes the efficient desorption and ionization of molecules from the vaporization of the matrix [57,58,59]. In general, this technology is used in clinical and research applications to study bacterial and fungal identification from a single colony [60], mutational identification, polymorphisms, insertion/deletion, splicing, quantitative changes variation, gene expression, and allele expression, as well as DNA methylation, and post-transcriptional modification of tRNAs and rRNAs [61,62].

One of the most exciting applications of MALDI is the analysis of the proteomic pattern composition of tumor cells and the determination of unique profiles that can actually differentiate normal cells from neoplastic cells, even between different subtypes of tumor cells and between primary and metastatic tumors, an approach that has already been explored in non-small cell lung cancer (NSCLC) [63]. SIMS, LDI, and MALDI are looking to minimize the analysis time of the imaging experiment.

### 6.5. Multiplexed Ion Beam Imaging and Imaging Mass Cytometry: The Antibody-Based Tag-Mass IMS Strategy

The tag-mass strategy is an affinity-based strategy where a probe, such as an antibody or an oligonucleotide, is directed against a specific target using a probe that can be imaged by any IMS strategy (Figure 2). The tag-mass needs a reporter group or element where the reporter is used to indirectly obtain the image from the probe attached to the target. The reporter must be designed to be an atom or molecule of known molecular mass that is easily detectable by MS, taking care to use a molecule that is not biologically active in the tissue to be analyzed. For SIMS or LA-ICP, metal isotopes conjugated to antibodies directly contain a monoatomic element that is easily detectable in biological studies. Where the element is not present naturally on the study surface, the former receives the name Multiplexed Ion Beam Imaging (MIBI) and the latter is named Imaging Mass Cytometry (IMC). The tag-mass IMS strategy is overwhelmingly expanding the possibilities of applications of mass-spectrometry to biological systems and biological samples. It is leading a revolutionary new wave of molecular and digital imaging, and it is the most powerful platform for multiplexing in the era of *theranostics*.

### 6.6. Multiplexed Ion Beam Imaging

Multiplexed ion beam imaging (MIBI) applies the principles of multi-isotope imaging mass spectrometry (MIMS) and the mass-tag strategy with metal-chelated isotopes conjugated to antibodies that will be incubated on the tissue of analysis (Figure 2) [64]. It allows subcellular imaging resolution. Instead of direct isotope labeling of the target cell or tissue, as described in the previous platforms, MIBI uses specific antibodies to “deliver” a specific mass to the targeted antigen. MIBI combines SIMS fundamentals, stable isotope reporters, specific antibodies, and intensive computation. 

MIBI is based on SIMS. An ionic beam erodes the surface or atomic layer of the sample, resulting in ionization of a small atomic fraction. In a SIMS instrument, a magnetic sector mass analyzer must filter the collected secondary ion beam; secondary ions are separated by mass and then used to derive a quantitative atomic mass image of the surface to be analyzed. Up to seven parallel masses of different elements or isotopes can be simultaneously analyzed, but by moving six of the seven detectors, the instrument can measure more data from multiple isotopes from the same region. The data are reconstructed into a grey scale image in which the pixel intensity is derived from the total number of counts of a given secondary ion within the area representing a given pixel (Figure 3). The lateral resolution is dependent on factors including the beam size and the number of pixels per image acquisition area [64].

Instead of fluorophores or enzyme-conjugated reagents, biological specimens for MIBI analysis are incubated with primary antibodies coupled to stable lanthanides that are highly enriched for a single isotope. The prepared specimens are mounted in a sample receptacle and subjected to a rasterized oxygen duoplasmatron primary ion beam. The coupling of MIBI to SIMS and TOF-MS allows the study of more than 50 metal-isotope labeled antibodies at the same time with a speed of 20–200 fields-of-view per day collected automatically. Depending on the element of interest, MIBI can achieve as low as parts-per-billion sensitivity with a dynamic range of 10^5^ and a resolution comparable to high-magnification light microscopy or close to 200 nm of resolution (Figure 1). MIBI is capable of analyzing standard FFPE tissue sections, fresh frozen tissue, and adherent cell samples [65,66]. The platform has a number of advantages over conventional multiplexed techniques, as there is no background because of the absence of autofluorescence and the very good definition of the signals for the image [67]. MIBI has been applied to demonstrate that it is a useful surrogate for standard IHC for diagnostics of molecular subtypes of breast cancer. In addition, through this platform, it could be possible to quantify the protein expression of markers [5,65], and as shown by Keren and colleagues [68] through the analysis of 36 proteins in 41 triple-negative breast cancer patients, the methodology has the capacity to provide data for application in immune oncology.

### 6.7. Imaging Mass Cytometry

Imaging Mass Cytometry (IMC) is a tag-mass IMS strategy coupled to Mass Cytometry by TOF (CyTOF). It uses LA-ICP and TOF MS fundamentals, in which antibodies are labeled with metal ion isotopes tags rather than fluorochromes [69] (Figure 1, Figure 2 and Figure 3). This allows the combination of many more antibody specificities (up to 50) in a single tissue sample (fresh frozen or FFPE) or adherent cell sample, without significant spillover between channels with a resolution from 1 μm up to 500 nm. Traditional labeling techniques can be used in this technique with minimal change to current protocols, allowing the panel design to be performed more easily and avoiding autofluorescence issues [70]. Although slower acquisition is observed (~1 h/mm^2^) and complete biological material is ablated, the IMC-CyTOF represents a new way of quantifying several phenotypes of cells at the same time, allowing the detection of up to 50 markers at the same time [71]. 

Currently, IMC has been used more often than MIBI in biological research. It is probably the most powerful platform for multiplex digital image analysis and together with MIBI, it is still undergoing development and improvement. IMC was developed on breast tumor tissue, and it is able to analyze cell-type markers, signaling activity, and hypoxia on FFPE samples, opening the possibility of carrying out deep analyses of tumor biology and heterogeneity [72]. Recently, the simultaneous multiplexed imaging of mRNA and protein expressions with subcellular resolution has been developed in breast cancer tissue samples by IMC, representing the first IMS platform that is able to study transcriptomic and protein expression with a high quality resolution, which expands the possibilities for applications to cancer research and overcomes the difficulties of the study of soluble proteins by conventional IHC [73].

## 7. Image Acquisition and Data Analysis

The main reason for performing a multiplexed assay is to obtain a high volume of tumor biological information through multidimensional data related to tissue architecture, spatial distribution of multiple cell phenotypes, co-expression of markers, and rare cell-type detection. The different advantages and disadvantages related to the study of multiple markers on a single slide are summarized in the Table 1, showing the methodologies described above. The first component after the staining is the image acquisition which must provide images with high enough quality to perform the analysis on. Currently the image acquisition systems are software-driven, robotically–controlled microscope systems that provide high quality monochrome cameras with high-resolution and multi-band filter cubes set to have greater flexibility and to match with the samples. Image acquisition systems alone or with their own analysis platform are used in the different methodologies described above, such as the Olympus scanner, Nikon Eclipse Ci-E [8], the Hamamatsu Nanozoomer S60 scanner with a Fluorescence Imaging Module [23], Zeiss/confocal laser scanning microscopy [22], Olympus America/VS110, Akoya/PerkinElmer Vectra^®^-Polaris^TM^ [30], Neo Genomics/MultiOmyx scanner [74], Ventana/Roche/iScan, Leica Biosystems/Aperio FL, 3Dhistech/Pannoramic/250 FLASH III [75], TissueGnostics/TissueFAXS [76], just to mention some. These have shown high versatility and quality imaging as well as the images generated by mass spectrometry techniques (Appendix A). Image acquisition systems for multiplexed immunofluorescence support multiple filters using mechanical switching or using tunable LED excitation, similar to the confocal microscope, to capture the fluorescence signals to assemble compose images [77] for analysis. The alignment of image acquired systems during successive rounds of staining is required for some staining techniques, such as MICSSS, SIMPLE, MELC, MultiOmyx^TM^, t-CyCIF, and CODEX, where it is essential to retain the information from each image during registration to stitch together images from overlapping fields to allow a precise representation of co-localization from different markers by the cells. Although the alignment of images is not necessary in other multiplexed methodologies because the image is acquired at the end of all staining process, it is still impossible to accelerate the process of scanning, which can take min to several hours [78] depending the size of the area scanned (whole section or region-of-interest, ROI), the number of ROIs scanned, and the methodology used [79,80]. In fluorescence methodologies, image acquisition is performed using one filter at a time or by changing the filter at each capture to obtain the co-localized [80] expression of the markers. The multidimensional tissue image generated by mass spectrometry techniques from the metal-labeled antibodies, such as MALDI-TOF, MIBI and CyTOF, that are used to perform highly multiplexed analyses are very comparable with the bright field or fluorescent images generated by the systems described above [81] (Figure 1). Overall, important considerations for cost estimation are the scan time, image resolution, hardware robustness, slide holder capacity, image focusing and stitching algorithms, acquisition modes, the use of bright field versus fluorescence, the file compression method/format/size, and the application capacity for these different techniques that need to be addressed, understood and discussed with the different vendors of image acquisition systems. The next component after the image acquisition is the image analysis and for that, several types of software have demonstrated their overall capability with different detection modules, including tissue segmentation, cell segmentation, co-localization, and spatial distribution of cell phenotyping, which is critically important to allow the combination of image layers to delineate the structures of interest to study multiplex staining tissues. A wide variety of image analysis software is applicable and is being developed to make high-dimensional image-based data exploration feasible for researchers who lack computational skills and flexible for computer scientists who want to develop and add advanced new methods for image-based machine learning-based phenotype scoring (Table 2). The combination of image analysis systems with automated scanning, such as Vectra^®^-Polaris^TM^/InForm Cell Analysis/Akoya/PerkinElmer [30], MultiOmyx/analysis software [74], Aperio FL/digital image analysis tools [82], is increasingly being employed to take advantage of multiplexed staining methodologies, all of which can scan slides affixed to whole tissue or tissue microarray slices prior to image analysis. Stand-alone image analysis software packages used to evaluate these virtual multiplex slides include Definiens TissueMap [83], HistoRx AQUA [84], SlidePath [85], Indica labs/HALO^™^ Image Analysis Software [86], and VISIOPHARM/Phenomap™ [87]. These are the most well-known software packages available in the market that offer high quality interpretation for multiplexed histological specimens. Open image analysis software, such as ImageJ/FIJI [88], QuPath [89], Icy [90] and Cell Profiler/Cell Analyst [91,92,93] are also available as open sources for multiplex image analysis with a high level of performance. In addition, it is important to mention that there are several companies that can provide different solutions for high content and/or high throughput scanning service and analytical algorithms, such as the TissueGnostics platform (https://www.news-medical.net/suppliers/TissueGnostics.aspx), Oncotopix^®^ (https://www.visiopharm.com/solutions/ oncotopix), 3DHISTECH Ltd. (https://www.3dhistech.com/quantcenter) and others such as Akoya/PerkinElmer (http://www.perkinelmer.com/corporate/what-we-do/markets/life-sciences/), Definiens (https://www.definiens.com/), Neogenomics (https://neogenomics.com/pharma-services/ lab-services/multiomyx), and IONpath (https://www.ionpath.com/) include a multiplex staining process with customized panels, scanning service and data analysis, and different strategies. High resolution performance during the multiplexing analysis across the ROIs/whole section needs to be combined with the signals of the immune markers to enable further different cell subpopulations to be identified and localized using the image analysis software. However, there are several types of image analysis software involving fluorescence and non-fluorescence multiplexed staining, as mentioned above, and their basic characteristics, such as having an easy algorithm workflow creation, and especially, having manual interactive or automated segmentation, with high flexibility cell phenotyping are important criteria to consider when choosing the image analysis system [94]. The power of different image analysis systems is reflected in the identification of cell phenotypes and in the specific pattern of immune cell identification, based either on the spatial distribution (distance between different subpopulations and cancer cells) or the relationships between different cells, such as lymphocyte subclasses, with each other (e.g., cytotoxic/regulatory cells) that can be associated with pathology, clinical patient information, and prognoses to give us important information about the tumor behavior [95].

However, although multiplex techniques are a powerful and efficient tool that allows us to identify several markers in a single slide, each methodology has a plethora of parameters that have significant effects on the outcome of the results and these need to be carefully validated in the lab, including antibody validation, tissue processing (cases and controls), signal acquisition calibration to obtain reproducible, reliable, and high-quality staining, and analysis that will be applied to clinical biopsies to provide a basic characterization of immune infiltrates to guide clinical decisions in the era of immunotherapy.

## 8. Clinical and Translational Use of Multiplexed Methodologies

Despite the evolution in previous years at different levels of cancer research concerning prevention, diagnosis, therapeutic options, and follow up methods, cancer still remains a major public health problem worldwide [100]. Immune contexture profiling is currently a powerful metric that can be used for tumor subclassification and the prediction of clinical outcomes [101]. A great variety of cancer research screening tools are applied to diagnose tumors, and these have been established for different tumors. Simultaneous quantification of more than one biomarker at the same time has become more and more interesting in cancer research using the technologies described previously [102]. Multiplexed methodologies can allow different biomarkers, representing different important systemic processes, such as inflammation, angiogenesis, or cell death, can be combined with established tumor markers in one single panel to potentially improve the study of cancer to aid in prevention, diagnostic accuracy, and treatment (Figure 4). Multiplex based immunoassays can offer important advantages, such as a high-throughput performance, low material requirement, a wide range of applications and cost- and time-effective multiplexing through the use of several parameters [23,96,103]. Several biomarkers could be cancer-specific, since malignant cells of different histologic types can produce different tumor-related patterns of proangiogenic factors, growth factors, and immune cells [68]. The study of biomarker panels can be used for early diagnosis and assessment of therapy responses [102]. The use of multiplexed methodologies to identify multiple biomarkers can be used to allow for the early detection of pre-neoplastic lesions, trying to identify basic microenvironment patterns on those cases to determine their progression to cancer [95] (Figure 5). Therefore, these new technology assays may represent an ideal method for developing personalized therapies if efficient multiplexing panels are created [4]. These technologies could help us to better understand the cancer microenvironment, highlighting the benefit for exploring immune evasion mechanisms and finding potential biomarkers that allow researchers to assess the mechanisms of action and predict and track responses [95].

## 9. Conclusions

Multiplexed methods can provide an important and efficient way to study disease diagnosis, prevention, and to carry out translational research. These systems are showing more and more different capabilities, from research labs towards the clinic, increasing the opportunity to better understand tumor–immune interactions. Multiplexed methodologies and image analysis strategies can allow important information about immune cell co-expression and their spatial-pattern distribution in the tumor microenvironment. However, the development of these new methods requires a multidisciplinary team including pathologists, oncologists, immunologists, engineers, and/or computer scientists. In addition, for research pathologists to use highly-multiplexed methods, these methodologies require automation to allow efficient and quick provision of information as well as easy analysis.

## Figures and Tables

**Figure 1 cancers-11-00247-f001:**
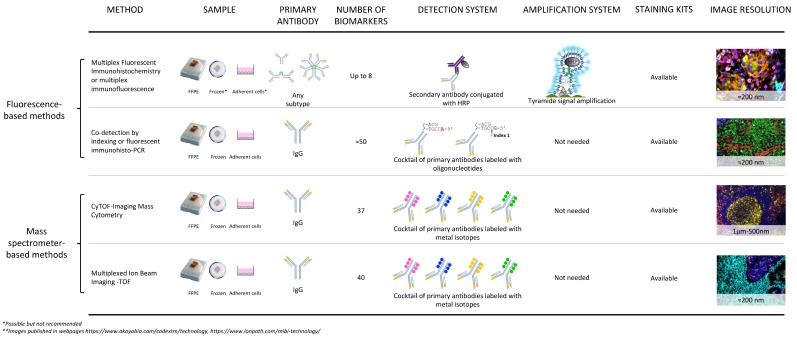
Technical comparison of fluorescent-based platforms and mass spectrometer-based platforms for digital image analysis. Digital image analysis for cancer research applications can be achieved with several methodologies. Some of them have advantages over others depending on the sample available, the specific antibodies against biological markers of interest, the detection system needed, and in some cases, the amplification of the signal for poorly expressed markers.

**Figure 2 cancers-11-00247-f002:**
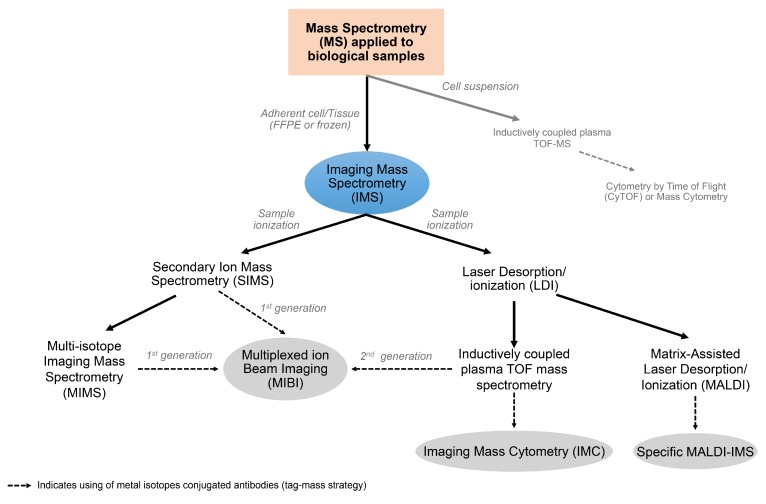
Byproducts and fundamentals of the imaging mass spectrometer. The application of mass spectrometry to biological research began in the last half century and it represents the conjunction of biological and deep physical and technological knowledge in biomedicine. Imaging Mass Spectrometry (IMS) came from the idea of building a 2D image with the elemental composition of a biological surface. The way that the surface is evaporated allowed the generation of two methods: one based on an ion beam and the second using a laser. The application of a tag-mass strategy to IMS is the most recent efficient and highly multiplexed platform for the digital image analysis of biological samples.

**Figure 3 cancers-11-00247-f003:**
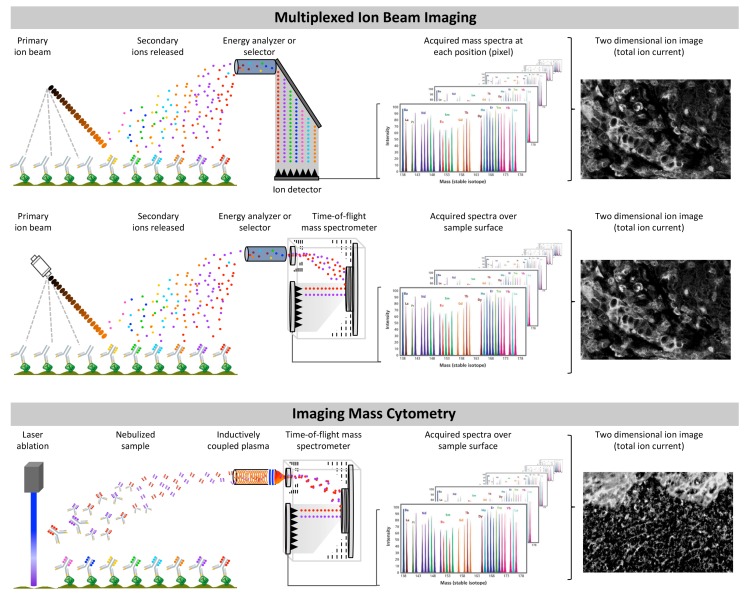
Basic fundamentals and similarities between multiplexed ion beam imaging and imaging mass cytometry. Characterization of multiplexed ion beam imaging and imaging mass cytometry.

**Figure 4 cancers-11-00247-f004:**
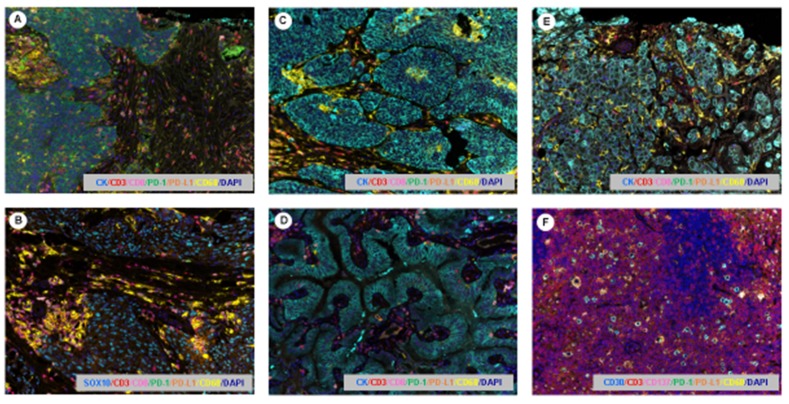
Multiplex immunoflorescencce microphotography. Images representing the immunoprofiling of different tumor types using the multiplexed tyramine signal amplification system: (**A**) esophageal squamous cell carcinoma, (**B**) malignant melanoma, (**C**) lung squamous cell carcinoma, (**D**) lung adenocarcinoma, (**E**) colorectal adenocarcinoma, (**F**) Hodgkin’s lymphoma. Scale bar: 200× magnification.

**Figure 5 cancers-11-00247-f005:**
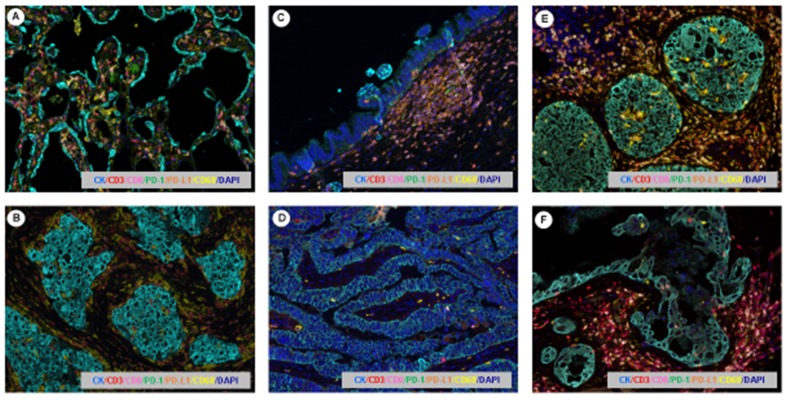
Multiplex immunofluorescence microphotography. Images representing the immunoprofiling of different stages of progression in lung, pancreas and breast cancer using the multiplexed tyramine signal amplification system: (**A**) pre-neoplastic lung lesion, (**B**) lung adenocarcinoma, (**C**) pre-neoplastic pancreatic lesion, (**D**) invasive pancreatic carcinoma, (**E**) non-invasive breast carcinoma, (**F**) invasive breast carcinoma. Scale bar: 200× magnification.

**Table 1 cancers-11-00247-t001:** Multiplex staining methodologies and their advantages and disadvantages.

Multiplex Staining Method	Advantage	Disadvantage
**Non-fluorescence based platform**		
Multiplexed immunohistochemical consecutive staining on a single slide	Uses conventional chromogenic-immunohistochemistry (IHC) staining Allows colocalization and detection of multiples proteins	Relatively slow process Request automatization Allows 10 labeled antibodies per slide
Sequential immunoperoxidase labeling and erasing	Use conventional chromogenic-IHC staining Allows colocalization and detection of multiple proteins Compatible with primary antibodies from same species	Relatively slow process Maximum of five antibody labels per section
**Fluorescence based platform**		
**Bleaching techniques without signal amplification system**		
Multi-epitope-ligand cartography	Allows colocalization and detection of a large number of proteins High functional resolution	The multiprobe image is limited to a single microscopic medium-to-high power field Longer sampling time The method requires robotic staining integrated with an inverted fluorescence microscope (high cost)
MultiOmyx^TM^ staining or hyperplexed Immunofluorescence Assay	Allows the analysis of up to 60 biomarkers in a single slide	Cycles of two antibodies with a longer sampling scan time
Tissue-based cyclic immunofluorescence method	Allows sequential immunostaining of around 30 markers	Slow process of around 6–8 h
Co-detection by indexing or fluorescent immunohisto-PCR	Eliminates autofluorescence Allows the analysis of several markers	Longer scan sampling time Limited use in formalin-fixed, paraffin-embedded (FFPE) tissues
DNA exchange imaging	Flexible for adaptation to diverse imaging platforms	Longer scan sampling time Small data analyzed
**Amplification of the epitope detection**		
Hapten-based modified multiplex	Fast staining around 2 h Cocktails of markers	Allows a maximum of 4 markers per slide Not tested with an autostainer
Tyramide signal amplification	Compatible with primary antibodies from the same species Available for autostainer	Allows a maximum of 7 labeled antibodies per slide
Nanocrystal quantum dots	Eliminates autofluorescence	Limited nanocrystals
**Mass Spectrometry Imaging**		
Secondary Ion Mass Spectrometry	The most sensitive system	The ionized particles destroy the region of interest (ROI) of analysis A current limitation is the availability of antibodies (high cost)
Laser Desorption/Ionization	Use of lasers (UV or IR) instead of ion beams High sensitivity and quantitative abilities	Low resolution
Matrix-assisted laser desorption/ionization	Organic compound matrix used	Sampling time and resolution A current limitation is the availability of antibodies (high cost)
Multiplexed ion beam imaging	Simultaneous labeling of up 40 antibodies with metals	Sampling time and small area sampling A current limitation is the availability of antibodies (high cost)
Imaging Mass Cytometry	Eliminates sample autofluorescence Preprocessing using routine immunohistochemistry protocols The signals are plotted using coordinates of each single laser shot No amplification step of the signal needed No matrix needed	Current limitations are the availability of antibodies (high cost), the sampling time, and the resolution Laser tissue ablation

**Table 2 cancers-11-00247-t002:** Image analysis software packages for multiplex staining.

Vendor	Software Package	Capabilities	Data Visualization	Availability	Reference
Akoya/PerkinElmer	InForm	Color-Based Co-localization, Tissue Segmentation, Cell/Object Segmentation, Cell Phenotyping, Scoring and Automated Quantitation using Batch Analysis	Density Raw Data	Licensed	[6,96]
Neo Genomics	MultiOmyx Quantification Program	Epithelial tissue reconstruction, Cellular and Subcellular Segmentation, Cell Phenotyping, Quantification Algorithms	Density Raw Data	Licensed	[3,20]
Leica Biosystems	Aperio eSlide Manager Analysis	Pixel-Based Analysis, Cellular identification, Area Quantification and Positive Pixel Count IF Algorithm	Density Raw Data	Licensed	[82]
Definiens	Tissue Studio/Image Developer	Imaging Segmentation, Marker Intensity Measurement, Cell Quantification, Batch Analysis, Statistical Analysis, and Algorithm Creator.	Histograms and Profile Plots	Licensed	[83]
HistoRx	AQUAnalysis	Signal Intensity Quantification Per Unit Area and Per Layer	Density Raw Data	Licensed	[84]
SlidePath	SlidePath’s Tissue Image Analysis	Membrane, Nuclear and Positive Pixel Quantification	Density Raw Data	Licensed	[85]
Indica Labs	HALO	Membrane, Co-localization, Immune Cell Proximity, Spatial Analysis, Batch Analysis	Spatial Plot, Histogram	Licensed	[86]
VISIOPHARM	Visimoph Tissuemorph	Signal Intensity, Area, Counting Objects, Spatial Analysis, Clustering Statistical Analysis, Batch Analysis and Algorithm Creator.	Phenotypic Matrix, t-SNE Plots	Licensed	[87]
Media Cybernetics	Image-Pro	Color-Based, Nuclear segmentation, Cell quantification, Macro-enabled Advanced Image Processing Solution	Density Raw Data	Licensed	[97]
CompuCyte	iCyte/iBroser/iNovator	Nucleus Segmentation or Phantom Contouring, Measures Associated Signals	Density Raw Data	Licensed	[98]
TissueGnostics	HistoQuest/TissueQuest/StrataQuest	Nuclei-Based Segmentation of Tissues, Cell Phenotyping	Density Raw Data	Licensed	[99]
NIH	Image J	Color-Based, User Interactive Segmentation	Histograms and Profile Plots	Open	[88]
https://qupath.github.io	QuPath	View Measurements in Context by Color Coding Objects According to Their Features, Flexible Object Classification, Trainable Cell Classification and Quantification	Density Raw Data	Open	[89]
http://icy.bioimageanalysis.org	Icy	Based and Color Object Identification, Size, Shape, Color Intensity, Texture, Spatial Analysis.	Plots, Histogram	Open	[90]
https://cellprofiler.org/	Cell Profiler/Cell Analyst	Based and Color Object Identification, Size, Shape, Color Intensity, Texture, and Number Neighbor Quantification.	Density Plot, Histogram	Open	[91,92,93]

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
