# Peer review of "State-of-the-Art of Profiling Immune Contexture in the Era of Multiplexed Staining and Digital Analysis to Study Paraffin Tumor Tissues"

_cancers, 2019, doi:10.3390/cancers11020247_

Round 1

Reviewer 1 Report

Again some typo in section "Image acquisition" :

"In florescence methodologies ..."

"Definians" => Definiens

"... image analysis software involving florescence...."

etc.

and also in Table 2: VISIOPHAM => VISIOPHARM

Author Response

Response to Reviewer #1:

Comment#1:

Again some typo in section "Image acquisition" :

"In florescence methodologies ..."

"Definians" => Definiens

"... image analysis software involving florescence...."

etc.

and also in Table 2: VISIOPHAM => VISIOPHARM

Response: We thank the reviewer for pointing out these typos. In the revised manuscript we correct these typos as well as in the Tables.

Reviewer 2 Report

I am satisfied with the amendments. 

Author Response

Response to Reviewer #2:

Comment#1:

I am satisfied with the amendments.

Response: Thank you to the reviewer.We feel that your suggestions increased the quality of this review.

This manuscript is a resubmission of an earlier submission. The following is a list of the peer review reports and author responses from that submission.

Round 1

Reviewer 1 Report

In this manuscript the authors review current methods of multiplex tissue staining, imaging, and image data analysis platforms in the context of quantification of the immune infiltrate. Up to the section titled: Image approaches and data analysis platforms, the review contains enough general information and details so that a reader who is already somewhat familiar with the compendium of existing technologies can follow. Figure 1 is informative. Summaries about mass spectrometry, ion beam imaging and imaging mass cytometry are satisfactory.  However, the reviewer does not find the same level of quality and details in the following sections.

First, the section: Image approaches and data analysis platforms lists most, but not all leading platforms available on the market. For example, TissueGnostics platform has been missed, just to give an example. In this section, ideally, all mentioned platforms should at least briefly be compared. Scanning speed, resolution of acquired images, number of biomarkers that can be imaged, scanning capacity: whole slides vs. ROIs, number of IF channels, etc, can be compared. The reason is to give readers an impression about what’s available on the market and what is best for one’s image data collections and numerical experiments that one can be interested in.

Second, 3D (confocal microscopy etc) imaging should be mentioned as an alternative to 2D in the context of immune infiltrate, its imaging and quantification. (Example literature https://www.ncbi.nlm.nih.gov/pmc/articles/PMC5360706/) Readers would certainly be interested in knowing where the 3D hardware and software techniques stand currently, and what 3D can offer versus 2D imaging in the context of immune infiltrate quantification.  Advantages (scanning depth) and disadvantages (scanning time, problems with antibody penetration of thick tissues) should at least be briefly specified. Here, as an example of a 3D imaging advantage, the authors could elaborate on issues of imaging of dendritic cells/macrophages (CD11b or CD11c positive). Can they be reliably imaged and quantitated in 2D?  Would 3D imaging be more suitable for this purpose?

Third, discussion on state-of-the-art commercial platforms and research-based solutions capable to quantitate immune infiltrate (with relevant examples from literature) should be incorporated into Image approaches and data analysis platforms and/or Clinical and translational use of multiplexed methodologies sections as well. Unfortunately, software that is a central component in cell enumeration pipelines has been completely missed in this review.

When adding relevant materials, the authors should also mention important measurements that are extracted and perhaps linked to clinical outcomes. For instance, cell subtypes in the immune infiltrate identified across the slide can be converted to cell density heatmaps. Furthermore, proximity analysis which includes but is not limited to measuring distances between T cells to tumor cells, or T cells to macrophages are important in understanding the level of interactions between cells and their activities in the tumor. With regards to the software platforms, the authors should search the literature to applications of commercial and examples of non-commercial solutions. https://onlinelibrary.wiley.com/doi/full/10.1002/path.5026, and Diagn Pathol. 2017 Sep 18;12(1):69 can serve as examples.

Author Response

Reviewer’s general comments and suggestions: We thank reviewer’s positive comments on our manuscript.

Response to Reviewer #1:

Reviewer #1: In this manuscript the authors review current methods of multiplex tissue staining, imaging, and image data analysis platforms in the context of quantification of the immune infiltrate. Up to the section titled: Image approaches and data analysis platforms, the review contains enough general information and details so that a reader who is already somewhat familiar with the compendium of existing technologies can follow. Figure 1 is informative. Summaries about mass spectrometry, ion beam imaging and imaging mass cytometry are satisfactory. However, the reviewer does not find the same level of quality and details in the following sections.

Comment#1: The section: Image approaches and data analysis platforms lists most, but not all leading platforms available on the market. For example, TissueGnostics platform has been missed, just to give an example. In this section, ideally, all mentioned platforms should at least briefly be compared. Scanning speed, resolution of acquired images, number of biomarkers that can be imaged, scanning capacity: whole slides vs. ROIs, number of IF channels, etc, can be compared. The reason is to give readers an impression about what’s available on the market and what is best for one’s image data collections and numerical experiments that one can be interested in.

Response: As suggested by the reviewer in the revised version of the manuscript we include the TissueGnostics platform and we created the Table 2 to do a briefly comparison of the most well know scanning software’s used to multiplexed assays.

Comment#2: 3D (confocal microscopy etc) imaging should be mentioned as an alternative to 2D in the context of immune infiltrate, its imaging and quantification. (Example literature https://www.ncbi.nlm.nih.gov/pmc/articles/PMC5360706/) Readers would certainly be interested in knowing where the 3D hardware and software techniques stand currently, and what 3D can offer versus 2D imaging in the context of immune infiltrate quantification. Advantages (scanning depth) and disadvantages (scanning time, problems with antibody penetration of thick tissues) should at least be briefly specified. Here, as an example of a 3D imaging advantage, the authors could elaborate on issues of imaging of dendritic cells/macrophages (CD11b or CD11c positive). Can they be reliably imaged and quantitated in 2D? Would 3D imaging be more suitable for this purpose?

Response: Thank you for your comment. Your comment is a good suggestion to compare in the future this two different technologies 3D as alternative methodology to 2D for immunoprofiling.

Comment#3: discussion on state-of-the-art commercial platforms and research-based solutions capable to quantitate immune infiltrate (with relevant examples from literature) should be incorporated into Image approaches and data analysis platforms and/or Clinical and translational use of multiplexed methodologies sections as well. Unfortunately, software that is a central component in cell enumeration pipelines has been completely missed in this review.

When adding relevant materials, the authors should also mention important measurements that are extracted and perhaps linked to clinical outcomes. For instance, cell subtypes in the immune infiltrate identified across the slide can be converted to cell density heatmaps. Furthermore, proximity analysis which includes but is not limited to measuring distances between T cells to tumor cells, or T cells to macrophages are important in understanding the level of interactions between cells and their activities in the tumor. With regards to the software platforms, the authors should search the literature to applications of commercial and examples of non-commercial solutions. https://onlinelibrary.wiley.com/doi/full/10.1002/path.5026, and Diagn Pathol. 2017 Sep 18;12(1):69 can serve as examples.

Response: As suggested by the reviewer in the revised manuscript we created a Table 3 listed the most available software to do image analysis describing their characteristics of analysis.

Reviewer 2 Report

Manuscript ID: cancers-398474

The authors propose an interesting review on multiplexed technologies to multiple epitope detection.

Several major comments on the manuscript:

1)    The authors erroneously included a multiplexed chromogenic immunohistochemistry approach, such as "SIMPLE", in a section on fluorescence-based platforms. The chromogenic/brightfield approach must be distinguished from the fluorescence-based one. In addition, more recent than SIMPLE approaches were developed and should be described (e.g., see Remark et al., Sci. Immunol 2016). Furthermore, other multiplexed chromogenic IHC approaches (without any bleaching steps) have demonstrated their usefulness and effectiveness in combination with the image analysis process for colour unmixing (see for example Ilie et al, Cancers 2018; Chen & Srinivas, Comput Med Imaging Graph. 2015). These chromogenic methods should be given particular attention because, unlike IF approaches, they easily allow morphological control, standardized processing of large tissue sample series, whole slide imaging and are more easily integrated into the routine clinical workflow of pathology departments.

2)    The section on digital analysis is very limited in the manuscript. Either the authors remove this reference from the title of the review, or (preferably) this section should be significantly expanded, describing the particularities versus similarities of the needs in terms of signal(/image) analysis for each technology. An important issue for signal analysis and relevant data extraction is the calibration of the signal acquisition technology and the control of irrelevant variations caused by the different marking techniques when several batches are required to analyze large clinical series (e.g. for biomarker validation). These controls are also necessary for a valid comparison of different series or studies and ultimately for clinical application. However, it seems that the authors are not very "comfortable" in this area because there are misinterpretations in the text. For example, Hamamatsu's NDPViewer is not an image analysis system but a simple and effective "viewer" for visual analysis and annotations of whole histological slide images. In addition, as this article focuses on multiplex technologies, reference 72 is not really relevant and should be replaced by the following reference from the same team. It concerns an image analysis method which is coupled to the "SIMPLE" technique and based on the open-source elastix framework: Moles Lopez X, et al J Am Med Inform Assoc. 2015 Jan;22(1):86-99.

I therefore strongly advise the authors to call upon a specialist on this issue to (at least review and preferably completely) redraft a relevant section on these image/signal analysis methods which are essential for the correct use of multiplex technologies.

3)    Because a laboratory would need to invest a significant amount of time and resources to establish multiplexed staining assays, this review should provide assistance to scientists and physicians to guide them in their choice between the different platforms. Guidelines and a summary table containing a series of criteria relating to the different technical aspects and constraints (e.g., number of markers, type (ROI or whole slide) of analysis regions, acquisition time, etc.), advantages and disadvantages (e.g., loss or no use of materials, cost, ease of implementation, ease of analysis, standardization and automation possibilities, etc.) should be provided to conclude this manuscript.

4)    Figures 1,3,4,5 included parts which are too small and/or of poor quality and thus difficult to read.

Minor remarks:

The “multidisciplinary team” mentioned in the conclusion should also includes engineers and/or computer scientists (cf. comment 2).

Author Response

Reviewer’s general comments and suggestions: We thank reviewer’s positive comments on our manuscript.

Response to Reviewer #2:

Reviewer #2: The authors propose an interesting review on multiplexed technologies to multiple epitope detection.

Major comments on the manuscript:

Comment#1: The authors erroneously included a multiplexed chromogenic immunohistochemistry approach, such as "SIMPLE", in a section on fluorescence-based platforms. The chromogenic/brightfield approach must be distinguished from the fluorescence-based one. In addition, more recent than SIMPLE approaches were developed and should be described (e.g., see Remark et al., Sci. Immunol 2016). Furthermore, other multiplexed chromogenic IHC approaches (without any bleaching steps) have demonstrated their usefulness and effectiveness in combination with the image analysis process for colour unmixing (see for example Ilie et al, Cancers 2018; Chen & Srinivas, Comput Med Imaging Graph. 2015). These chromogenic methods should be given particular attention because, unlike IF approaches, they easily allow morphological control, standardized processing of large tissue sample series, whole slide imaging and are more easily integrated into the routine clinical workflow of pathology departments.

Response: We thank the reviewer for pointing out this mistake. In the revised manuscript, we created a new section “Non-fluorescence-based platforms” and we described this methodology “SIMPLE” in that section, page 4 as well as we introduce other multiplexed chromogenic IHC approach as “Multiplexed immunohistochemical consecutive staining on single slide (MICSSS)”, Page 4. As followed:

“The Multiplexed immunohistochemical consecutive staining on single slide (MICSSS) (8) method is a sequential cycles of staining, image scanning, and destaining of chromogenic substrate than can be performed on FFPE tissue samples. This multiplex staining approach use conventional chromogenic-immunohistochemistry staining followed after the scanning process by the destained in organic solvent (8) that can remove completely the staining. The MICSS method can allow up 10 different antibodies on one single tissue section using sequential cycles without any damage to the tissue antigenicity or architecture. The relatively slow process of the technique is the main limitation of the MICSSS but as the authors mentioned this limitation can be easily resolve with the automation of the process. However, this methodology was tested in a limited data, it showed their versatility and potential to study and analyze the complexity of the tumor microenvironment”.

Comment#2: The section on digital analysis is very limited in the manuscript. Either the authors remove this reference from the title of the review, or (preferably) this section should be significantly expanded, describing the particularities versus similarities of the needs in terms of signal (/image) analysis for each technology. An important issue for signal analysis and relevant data extraction is the calibration of the signal acquisition technology and the control of irrelevant variations caused by the different marking techniques when several batches are required to analyze large clinical series (e.g. for biomarker validation). These controls are also necessary for a valid comparison of different series or studies and ultimately for clinical application. However, it seems that the authors are not very "comfortable" in this area because there are misinterpretations in the text. For example, Hamamatsu's NDPViewer is not an image analysis system but a simple and effective "viewer" for visual analysis and annotations of whole histological slide images. In addition, as this article focuses on multiplex technologies, reference 72 is not really relevant and should be replaced by the following reference from the same team. It concerns an image analysis method which is coupled to the "SIMPLE" technique and based on the open-source elastix framework: Moles Lopez X, et al J Am Med Inform Assoc. 2015 Jan;22(1):86-99.

I therefore strongly advise the authors to call upon a specialist on this issue to (at least review and preferably completely) redraft a relevant section on these image/signal analysis methods which are essential for the correct use of multiplex technologies.

Response: As suggested by the reviewer we expand and re-wrote the section “Image approaches and data analysis platforms” in the revised version of the manuscript, page 18 (see below). In addition we created 3 new tables, doing the comparison of the different techniques for multiplexed (Table 1), described the multiplex imaging scanning products (Table 2), and described the image analysis software packages for multiplex analysis (Table3). Also, in the revised version of the manuscript we clarify that the Hamamatsu's NDPViewer is not an image analysis software.

Image approaches and data analysis platforms

“Although, multiplexed staining available for FFPE or frozen material enables multi-parametric readouts from a single tissue section, the different techniques described before have advantage and disadvantages related to the technique (Table 1), as well as sometimes limited scalability and throughput, related to limited small region-of-interests (ROI) scanning or limited to few number of fields-of-views (74, 75). The major part of the scanner system provided high quality of monochrome cameras with high-resolution and multi-band filter cubes set that provided greater flexibility, to match with the sample. The improvement of the image acquisition scanners provide in this new era high quality of image resolution for multiplex staining, different scanner alone and others with their own image analysis software used in some of the fluorescence and non- fluorescence methodologies mentioned above as Olympus scanner (OlyVIA software), Nikon Eclipse Ci-E microscope (8), Hamamatsu Nanozoomer S60 scanner with a Fluorescence Imaging Module (23), confocal laser scanning microscopy(22), Vectra®-PolarisTM (30), MultiOmyx (76), Bacus TMAScore, Dako ACIS III, Genetix Ariol, Aperio FL, 3DHistech Mirax HistoQuant (77), showed high versatility and high quality of image acquisition, Table 2. Scanners using for multiplexed immunofluorescence staining slides need to support multiple filters using mechanical switching or using tunable LED excitation, similar to confocal microscope, to capture the fluorescence signals and assemble in a compose image (78), ready for the analysis procedure. Although, the fluorescence scanner systems can capture the ROIs using one filter at the time or changes the filter at each capture to high channel of co-localization (75), is still impossible to accelerate the process of the scanning to obtain high quality of images in this instruments and it is variable depending the methodology used in the scanning that can takes from minutes to serval hours (79). In addition, the multidimensional tissue image generated by the different mass spectrometry techniques from the metal-labeled antibodies, as MALDI-TOF, MIBI and CyTOF, are very comparable with the bright field or fluorescence images generated by the different scanner to perform highly multiplexed analysis (80), Figure 1. In this regard, image analysis software’s need to be accessible for the different techniques, easy with automated capabilities of detection, including tissue segmentation and spatial co-localization cell distribution, critically important to study in particular small samples, such as core needle biopsies or small metastatic tumor samples. A wide variety of image analysis software approaches are applicable and in developing to make high-dimensional image-based data exploration feasible for researchers who lack computational skills and flexible for computer scientists who want to develop and add advanced new methods for image-based machine learning-based phenotype scoring, Table 3. Image analysis systems combined with automated scanning, are increasingly being employed to take advantages of these multiplexed staining methodologies as Vectra®-PolarisTM/InForm Cell Analysis/PerkinElmer/Akoya (30), MultiOmyx/analysis software (76), Aperio FL/digital image analysis tools, 3DHistech Mirax HistoQuant, (77), all of which can scan slides affixed to whole tissue or tissue microarray slices prior to image analysis. Stand-alone image analysis software packages used to evaluate these virtual multiplex slides include Definiens TissueMap, HistoRx AQUA, SlidePath (81), CRi Nuance (82), Indica labs/HALO Image Analysis Software, and VISIOPHARM/Phenomap™ are the most know software packages available in the market that offer high level of quality interpretation for multiplexed histological specimens. In addition, open image analysis software’s as ImageJ/FIJI (83), QuPath (84), Cell Profiler/Cell Analyst (85, 86) and image analysis platforms as TissueGnostics platform, to name a few, also guarantee high performance analysis. Whole tissue scanner software viewers as Hamamatsu NDPViewer (77, 87) are also important to help during the scanning and analysis process. High resolution performance during the multiplexing analysis across the ROIs/whole section need to be combined with the signals of the immune markers that will be enabled further different cell subpopulations to be identified and localized using the image analysis software’s. However, there are several image analysis software’s to florescence and non-fluorescence multiplexed staining as mentioned above, basic characteristics as easy algorithm workflow creation, and especial, manual interactive or automated segmentation, with high flexibility of cells phenotyping will be important criteria’s to choose the image analysis system (88). The power of different image analysis systems will be reflecting in the identification of cell phenotypes’ and in the specific pattern of immune cell identification, based either on the spatial distribution (distance between different subpopulation and cancer cells) or the relationships of different cells, as lymphocyte subclasses, with each other (e.g. cytotoxic/regulatory cells) that can be associated with pathologic and clinical patient information and prognosis, to give us an important information about the tumor behavior (89). However, the multiplex techniques are a powerful and efficient tool that allows us to identify several markers in one single slide, each methodology have a plethora of parameters that have significant effects on the outcome of the results and need to be carefully validated in the lab, including antibody validation, tissue process (cases and controls), and signal acquisition calibration to obtain a reproducible, reliable, and high-quality staining data and methodology that will be applied  to clinical biopsies to provide a basic characterization of immune infiltrates to guide clinical decisions in the era of immunotherapy”.

Comment#3: Because a laboratory would need to invest a significant amount of time and resources to establish multiplexed staining assays, this review should provide assistance to scientists and physicians to guide them in their choice between the different platforms. Guidelines and a summary table containing a series of criteria relating to the different technical aspects and constraints (e.g., number of markers, type (ROI or whole slide) of analysis regions, acquisition time, etc.), advantages and disadvantages (e.g., loss or no use of materials, cost, ease of implementation, ease of analysis, standardization and automation possibilities, etc.) should be provided to conclude this manuscript.

Response: As suggest by the reviewer in the revised version of the manuscript we created an additional Table 1, 2 and 3 to showing the characteristics of the multiplexed techniques, scanner systems and analysis software’s.

Comment#4: Figures 1,3,4,5 included parts which are too small and/or of poor quality and thus difficult to read.

Response: In the revised version of the manuscript we improve the quality of the figures 1,3,4,5. Please note that the images that are included directly in the manuscript some time loss their resolution for this reason we sent to the journal a high quality PDF images in a different file that the review can open.

Minor remarks:

Comment#1: The “multidisciplinary team” mentioned in the conclusion should also include engineers and/or computer scientists (cf. comment 2).

Response: As suggest for the review in the revised version of the manuscript we include in the multidisciplinary team engineers and/or computer scientists.

Reviewer 3 Report

The authors present here a comprehensive review on the various multiplexed technologies available for high dimensional digital pathology. The amount of work and research done is very extensive, and this review will be very important for pathologists and biologists venturing into multiplexed imaging methods.

I would like to suggest a couple of areas, mostly technical, which the authors can consider to improve this comprehensive review:

1. The authors did a very admirable job of describing the various multiplexed tissue staining methods. Of note, CyCIF from Peter Sorger's lab (Lin et al 2018 ELife) and DEI from Peng Yin's group (Wang et al 2017 Nano Letters) are recent methodogical advancements that should be touched upon. They have theoretical multiplexibility of similar levels as CODEX, although the implementation methods are very different.

2. Figure 1 can be misleading, I would advise double checking the numbers. Most multiplexed FFPE slide images currently are acquired/scanned at 20X, which is ~400-500nm XY resolution. For MIBI, down to 260nm resolution can be acquired, but ~500nm resolution (equivalent to 20X) was used for Keren et al 2018 on the MIBI-ToF. Hence, I would suggest changing the "Image Resolution" to "Practical Image Resolution", to reflect the general image quality captured (instead of best possible, which will take a significantly longer time). 

3. The authors should consider citing Keren et al 2018, which highlights the MIBI-ToF and the first set of data acquired on it. This is a custom built instrument based off the initial proof of concept of MIBI performed on a NanoSIMS.

 Minor typos/grammatic errors. Eg

 Line 25: "showed *his* utility"

 Line 377: "as the high frequent as the low frequent"

 Line 390: "cancer diseases still the major public health problem worldwide"

 Thank you for taking the time to write this extensive review! I did enjoy reading about the overview of method development.

Author Response

Reviewer’s general comments and suggestions: We thank reviewer’s positive comments on our manuscript.

Response to Reviewer #3:

Reviewer #3: The authors present here a comprehensive review on the various multiplexed technologies available for high dimensional digital pathology. The amount of work and research done is very extensive, and this review will be very important for pathologists and biologists venturing into multiplexed imaging methods.

I would like to suggest a couple of areas, mostly technical, which the authors can consider to improve this comprehensive review:

Comment#1:  The authors did a very admirable job of describing the various multiplexed tissue staining methods. Of note, CyCIF from Peter Sorger's lab (Lin et al 2018 ELife) and DEI from Peng Yin's group (Wang et al 2017 Nano Letters) are recent methodogical advancements that should be touched upon. They have theoretical multiplexibility of similar levels as CODEX, although the implementation methods are very different.

Response: As suggested by the reviewer in the revised version of the manuscript we mentioned these two new recent technologist that have theoretical multiplexibility of similar levels as CODEX in the section, Tissue-based cyclic immunofluorescence (t-CyCIF) method, Page 7 as followed: “Recently described in the literature the tissue-based cyclic immunofluorescence (t-CyCIF) (22) method can creates a highly multiplexed images using an iterative process in which conventional low-plex fluorescence images are repeatedly collected from the same sample and then assembled into a high-dimensional representation. The t-CyCIF cycles involve the antibody staining against protein antigens, nuclear staining (same fluorophore per cycle), image scanning (low and high magnification) and fluorophore bleaching step, the cycles can be repeated according the authors more than 15 times without any problem with the cells preservation or tissue morphology, to complete all the desired targets. However, each t-CyCIF cycle involves a relatively slow process (each cycle 6–8 hr), a single operator can process 30 slides in parallel with relatively flexibility. Recently, Bolognesi and colleagues (23) described a multiplex staining by sequential immunostaining involving 30 markers using Beta-mercaptoethanol/sodium dodecyl sulfate as stripping procedure during each cycle of staining and scanning with very good results”.

            As well as in the section, Co-detection by indexing or fluorescent immunohisto-PCR, Page 8 we described the DNA exchange imaging (DEI) technique, as followed: “Another barcoding platform is the DNA exchange imaging (DEI) technique (25) that overcomes speed restrictions by allowing for single-round immunostaining with DNA-barcoded antibodies. The DEI is the new generation of exchange-PAINT described by the same group (26), according the authors is an easy multiplexed technique than can be adaptable to diverse imaging platforms, including standard resolution Exchange-Confocal and various super-resolution methods. However, there is no cancer-related study publications using these methods, but those are promising techniques and highly efficient methodology to study the tumor-associated immune contexture”.

Comment#2: Figure 1 can be misleading, I would advise double checking the numbers. Most multiplexed FFPE slide images currently are acquired/scanned at 20X, which is ~400-500nm XY resolution. For MIBI, down to 260nm resolution can be acquired, but ~500nm resolution (equivalent to 20X) was used for Keren et al 2018 on the MIBI-ToF. Hence, I would suggest changing the "Image Resolution" to "Practical Image Resolution", to reflect the general image quality captured (instead of best possible, which will take a significantly longer time).

Response: Thank you to the reviewer to check the misleading in Figure 1. In the revised version of the paper we corrected it.

Comment#3: The authors should consider citing Keren et al 2018, which highlights the MIBI-ToF and the first set of data acquired on it. This is a custom built instrument based off the initial proof of concept of MIBI performed on a NanoSIMS.

Response: In the revised version of the manuscript we include Karen and colleagues as the first set of data acquired by the MIBI-ToF, section Multiplexed ion beam imaging, Page 17 “as showed by Keren and colleagues (68), with the analysis of 36 proteins in 41 triple-negative breast cancer patients the capacity of the methodology to provide data for their application in the immune oncology”.

Comment#4: Minor typos/grammatic errors. Eg

 Line 25: "showed *his* utility"

 Line 377: "as the high frequent as the low frequent"

 Line 390: "cancer diseases still the major public health problem worldwide"

Response:

We revised carefully the manuscript and we fixed the errors in the revised version.

Round 2

Reviewer 1 Report

The newly introduced section on image analysis is very general and lacks specifics of software capabilities that differentiate one package from another. In the previous round, the reviewer pointed out some analytical software that were not addressed.  This was pointed out by another reviewer as well.  Table 2 should be moved to a supplement.

The table where the software is listed should be supplemented with references and the manuscript should be amended with the context/purpose select packages were used for. It seems that the authors confuse image analysis software with software packages that operate the scanners. The products of TissueGnostics are not listed in tables with scanning and image analysis platform. Is there a reason why?

English needs to be polished and checked for grammar and phraseology.

Author Response

Response to Reviewer #1:

Comment#1:

The newly introduced section on image analysis is very general and lacks specifics of software capabilities that differentiate one package from another. In the previous round, the reviewer pointed out some analytical software that were not addressed.  This was pointed out by another reviewer as well.  Table 2 should be moved to a supplement.

Response: As suggested by the reviewer we addressed the capabilities of the software’s in the Table 2. In addition, in the new version of the manuscript we mentioned companies that offer analytical software’s and solution for this type of image analysis. The Table 2 was changed as supplementary Table 1.

Comment#2:

The table where the software is listed should be supplemented with references and the manuscript should be amended with the context/purpose select packages were used for. It seems that the authors confuse image analysis software with software packages that operate the scanners. The products of TissueGnostics are not listed in tables with scanning and image analysis platform. Is there a reason why?

Response: As suggested by the reviewer we added references in the Table 2 of the revised manuscript and we mentioned about TissueGnostics in the Tables as well as other companies that offer similar solutions of different products.

Comment#3:

English needs to be polished and checked for grammar and phraseology.

Response: The new version of the manuscript pasted for an extensive English edition. 

Reviewer 2 Report

I am not satisfied with the new "image analysis" section, from which it is very difficult to extract useful information. Moreover, the writing seems sloppy: the sentences are poorly structured, there are typographical errors...

It is necessary for authors to take the time to review this section and improve its writing and structure to make it useful to the reader.

In addition, some of the requested information is missing, such as specific image processing needs for some multiplexing approaches compared to others. For example, some approaches require image alignment/registration to judge the co-expression of several proteins by cells, while others do not.

In Table 2 there is a mixture between imaging devices (scanner, etc.) and software. There is also inconsistency with the title of the "program" column, whereas most of the references are scanners, mixed with some software, such as OlyVIA, AxioVision, etc. (which should be coupled to a compatible image acquisition device).

Author Response

Response to Reviewer #2:

I am not satisfied with the new "image analysis" section, from which it is very difficult to extract useful information. Moreover, the writing seems sloppy: the sentences are poorly structured, there are typographical errors...

Comment#1:

It is necessary for authors to take the time to review this section and improve its writing and structure to make it useful to the reader.

Response: As suggested by the reviewer the “Image acquisition and data analysis” section pasted for an extensive revision.

Comment#2:

In addition, some of the requested information is missing, such as specific image processing needs for some multiplexing approaches compared to others. For example, some approaches require image alignment/registration to judge the co-expression of several proteins by cells, while others do not.

Response: As suggested by the reviewer we added some missing information as image alignment for co-expression analysis required with some stained platforms during the image acquisition in the revision version of the manuscript.

Comment#3:

In Table 2 there is a mixture between imaging devices (scanner, etc.) and software. There is also inconsistency with the title of the "program" column, whereas most of the references are scanners, mixed with some software, such as OlyVIA, AxioVision, etc. (which should be coupled to a compatible image acquisition device).

Response: We thank the reviewer for pointing out this mistakes. In the revised manuscript in the supplementary table 1 we excluded the software’s related with scanner and we mentioned only image acquisition instruments.